# DIFFUSION-GUIDED SAFE POLICY OPTIMIZATION FROM COST-LABEL-FREE OFFLINE DATASET

## ABSTRACT

Offline safe reinforcement learning (RL) aims to guarantee the safety of decision-making in both training and deployment phases by learning the safe policy entirely from offline data without further interaction with the environment, which pushes the RL towards real-world applications. Previous efforts in offline safe RL typically presume the presence of Markovian costs within the dataset. However, the design of a Markovian cost function involves rehearsal of all potentially unsafe cases, which is inefficient and even unfeasible in many practical tasks. In this work, we take a further step forward by learning a safe policy from an offline dataset without any cost labels, but with a small number of safe demonstrations included. To solve this problem, we propose a two-stage optimization method called **D**iffusion-guided **S**afe **P**olicy **O**ptimization (**DSPO**). Initially, we derive trajectory-wise safety signals by training a return-agnostic discriminator. Subsequently, we train a conditional diffusion model that generates trajectories conditioned both on the trajectory return and the safety signal. Remarkably, the trajectories generated by our diffusion model not only yield high returns but also comply with the safety signals, from which we can derive a desirable policy through behavior cloning (BC). The evaluation experiments conducted across tasks from the SafetyGym, BulletGym, and MetaDrive environments demonstrate that our approach can achieve a safe policy with high returns, significantly outperforming various established baselines.

## 1 INTRODUCTION

Reinforcement Learning (RL) has demonstrated remarkable potential and efficacy across a wide range of applications, from gaming (Berner et al., 2019) and robotics (Singh et al., 2022) to financial decision-making (Wang et al., 2019) and healthcare (Coronato et al., 2020). It promises to build effective agents in various decision-making domains. However, real-world applications are more complicated, with certain constraints for the deployed decision-making policy. One of these requirements is the safety of the RL policy. To satisfy this requirement, safe reinforcement learning aims to optimize the agent policy with an explicit safety constraint, thus maintaining the decision making safety while improving the task performance. However, conventional safe RL approaches (Achiam et al., 2017; Stooke et al., 2020; Liu et al., 2022), similar to the standard RL, require a large amount of online interaction with the environment for policy training. This practice can lead to unsafe interactive behaviors during training because the RL algorithm requires exploration to learn a good policy.

To achieve a safe training process instead of only safe policy deployment, offline safe RL (Liu et al., 2023; Xu et al., 2022b; Lee et al., 2022) proposes to learn safe policy entirely on the offline dataset without any necessity for extra interaction with the environment. BCQ-Lag (Xu et al., 2022b) and BEAR-Lag (Xu et al., 2022b) incorporate the Lagrangian method with existing offline RL algorithms to penalize constraint violations. CPO (Xu et al., 2022b) proposes a policy search algorithm with approximate theoretical guarantees for constraint satisfaction at each iteration. CDT (Liu et al., 2023) embeds the safety constraint into the decision transformer to model constraint sequences. By avoiding unsafe training, these recent works push the safe RL further toward real-world applications.

Despite the progress, all these previous offline safe RL works assume that there exist transition-level cost labels in the dataset (Liu et al., 2024; Xu et al., 2022b; Liu et al., 2023), which may not hold in many real-world problems. On one hand, for a cost function to be comprehensive, it needs to account for all potential unsafe cases to prevent cost hacking, a process that tends to be notably

inefficient in complex tasks. On the other hand, safety sometimes requires consideration of entire trajectories rather than individual transitions, because capturing all task-relevant information within a single state representation is difficult (Bacchus et al., 1996; 1997; Kim et al., 2022). For example, in one case where a relatively fast-moving car fails to navigate a V-shaped turn and collides with the guardrail, it is challenging to define the cost associated with each action taken during its prior normal driving. Despite the challenges in defining a Markovian cost function, acquiring a small set of safety demonstrations is often feasible in many scenarios (Fang et al., 2019; Le Mero et al., 2022; Li et al., 2022a), e.g., it is possible to obtain a few demonstrations of safe and qualified driving. Therefore, we propose a new problem setup in which the agents are expected to learn safe policies only from an offline dataset without any cost labels, but a small number of safe demonstrations are provided.

To solve this problem setup, we propose a **D**iffusion-guided **S**afe **P**olicy **O**ptimization method (**DSPO**), where we firstly build a trajectory-wise safety discriminator to determine whether one trajectory is safe or not, and then utilize a conditional diffusion model to help derive the safe policy. More specifically, inspired by Chen et al. (2021); Kim et al. (2022), we employ a transformer-based discriminator, called SafetyTransformer, to encode task-related safety information from trajectories. SafetyTransformer processes trajectory inputs and generates trajectory-level safety signals, treating unlabeled demonstrations as negative examples and limited safe demonstrations as positive examples during training. Additionally, as demonstrations in the safe dataset often produce high returns, conventional discriminator learning approaches may assign high weights to some unsafe yet high-return trajectories. To address this issue by learning return-agnostic safety signals, we introduce an extra training objective aimed at minimizing the mutual information (MI) between the output logits of the SafetyTransformer and the trajectory returns. Following this, we label all the trajectories in the dataset with the learned safety signals to train a conditional diffusion model that takes trajectory return and the labeled safety signal as conditions. Subsequently, we utilize the diffusion model to generate safe qualified trajectories with high returns from which we can derive a desirable policy through behavior cloning (BC).

In the experimental part, we build an offline dataset suite including tasks from SafetyGym (Ji et al., 2023), BulletGym (Gronauer, 2022), and MetaDrive (Li et al., 2022b) environments, where exists a set of offline trajectories and a few number of safe demonstrations for each task. The extensive experiments demonstrate that our approach can obtain a safe policy with high returns in this more challenging setting, significantly outperforming various baselines. The main contributions of this work are summarized as follows:

- We propose a problem setup that is more practical in certain scenarios, pushing safe reinforcement learning one step forward toward real-world applications.

- We propose a novel approach for the proposed setup which builds a trajectory-wise safety discriminator to determine whether one trajectory is safe or not, and then utilize a conditional diffusion model to help derive the safe policy.

- Building an offline dataset suite across diverse tasks, we have benchmarked various types of algorithms under our proposed problem setting. Extensive experiments show that our approach outperforms other baselines in this problem setup.

## 2 PROBLEM FORMULATION

In the field of Reinforcement Learning (RL), each task can be effectively modeled as a Markov Decision Process (MDP), denoted by $\mathcal{M} := \langle \mathcal{S}, \mathcal{A}, \mathcal{P}, \mathcal{R}, \gamma, \rho_0 \rangle$. Here, $\mathcal{S}$ and $\mathcal{A}$ represent the state space and action space respectively, $\gamma \in [0, 1)$ is the discount factor, and $\rho_0$ is the initial state distribution. At every timestep $t$, an agent selects an action $a_t \in \mathcal{A}$, causing the environment to transition to a subsequent state $s_{t+1} \in \mathcal{S}$, guided by the transition function $\mathcal{P}(s_{t+1}|s_t, a_t)$. Simultaneously, the agent receives a reward $r_t = \mathcal{R}(s_t, a_t)$, which provides feedback for the action taken. The primary goal of the agent is to maximize the expected discounted return, formally expressed as

$$\mathbb{E}_{s_0 \sim \rho_0, s_{t+1} \sim \mathcal{P}(\cdot|s_t, a_t), a_t \sim \pi(\cdot|s_t)} \left[ \sum_{t=0}^{\infty} \gamma^t \mathcal{R}(s_t, a_t) \right],$$

where the sum of discounted rewards over an infinite horizon is considered.

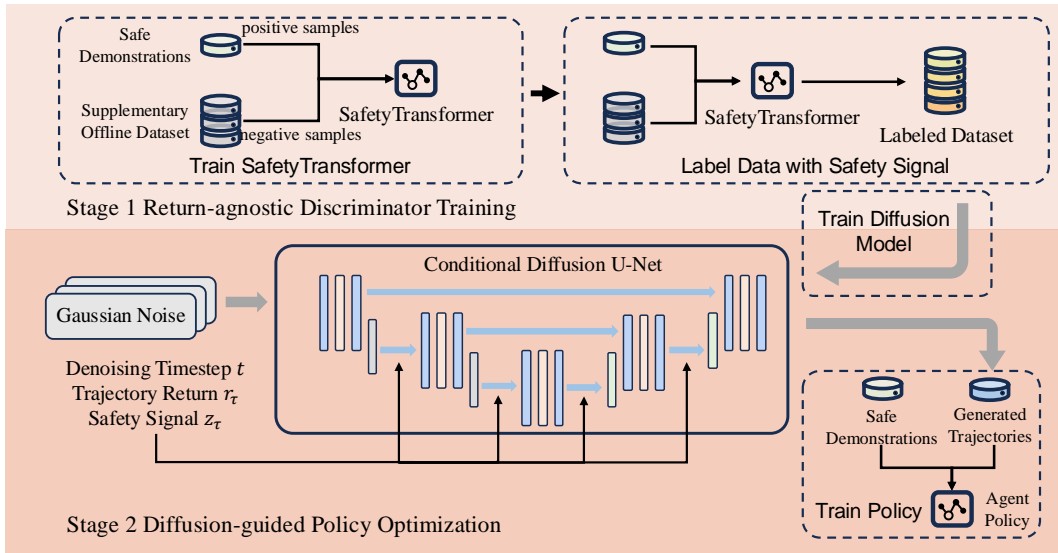

Figure 1: The whole framework of our proposed approach. In Stage 1, we leverage safe demonstrations and supplementary data to train the SafetyTransformer, which labels the dataset with safety signals. In Stage 2, we train a conditional diffusion model on the labeled dataset and distill the learned safety knowledge into the agent policy using behavior cloning.

For offline RL, instead of interacting with the environment, the agent can only learn policy from a fixed offline dataset $\mathcal{D}$, where each $\tau \in \mathcal{D}$ denotes a trajectory $\{s_1, a_1, r_1, s_2, a_2, r_2, \ldots, s_T, a_T, r_T\}$. Previous works on offline safe RL typically include Markovian cost labels within these trajectories, assigning a cost value $c_t$ analogous to $r_t$ for each transition, facilitating the enforcement of safety constraints. In contrast, our problem setup diverges by not presupposing the presence of Markovian costs, where each trajectory is solely composed of sequences of states, actions, and rewards like standard offline RL.

However, a significant *assumption* is that the offline dataset consists of two parts: a small number of safe demonstrations $\mathcal{D}^S = \{\tau_i^S\}_{N^S}$ and another supplementary offline dataset $\mathcal{D}^U = \{\tau_i^U\}_{N^U}$ whose safety is unknown. In most cases, the number of trajectories in the safe demonstrations, i.e., $N^S$, is much smaller than that of the supplementary offline dataset $\mathcal{D}^U$, i.e., $N^U$. In this problem setup, the agent is expected to learn safe policy solely from the datasets $\mathcal{D}^S$ and $\mathcal{D}^U$ without further interaction with the environment.

## 3 METHOD

To learn safe policies from offline data without any cost labels, we propose a two-stage optimization method called **D**iffusion-guided **S**afe **P**olicy **O**ptimization (**DSPO**). In the first stage, we train a transformer-architecture discriminator to derive trajectory-wise safety signals, which is designed to be return-agnostic. Then in the second stage, the established safety signals are utilized to train a conditional diffusion model, which guides the safe policy optimization through generating safe trajectories with high returns. Figure 1 depicts the whole framework of our approach. We also provide an algorithm pseudo-code in Appendix D.2 to show the overall process of our approach.

In fact, directly applying offline RL techniques is infeasible in our problem setup, because the safety constraints are not explicitly represented in the offline dataset. That is to say, some high return trajectories may represent unsafe behavior patterns, misleading offline RL methods into deriving unsafe policies. To solve this issue, we propose to learn safety signals that help determine the safety weight of each trajectory Considering the Markovian safety signal is not applicable to all scenarios, we propose to learn the safety signal at the trajectory level by utilizing a transformer-architecture discriminator, SafetyTransformer. Moreover, to avoid assigning high weights to unsafe trajectories yet with high returns, we propose to learn SafetyTransformer in a return-agnostic manner. All details concerning the discriminator training are provided in Section 3.1.

Upon establishing the safety signals, we transformed the original dataset into trajectories that include labels related to both task performance and safety. Our goal then shifts to optimizing a safe policy by utilizing the return and safety information from each trajectory. The core challenge is to understand the behavior patterns under varying trajectory returns and safety signals. This task is non-trivial because the trajectory distribution is influenced by two objective variables, making it difficult to capture accurately. To address this challenge, we propose using a conditional diffusion model to fit the trajectory distribution, inspired by the diffusion model's remarkable expressive capability. However, since the diffusion model cannot directly output decision actions and instead expresses the joint distribution of an entire trajectory, we further distill a policy by generating safe, qualified trajectories using the diffusion model and performing behavior cloning (BC). More details about diffusion-guided safe policy optimization can be found in Section 3.2.

## 3.1 Trajectory-wise Safety Signal

Previous offline safe RL works typically assume the presence of Markovian cost labels within the dataset, which are not provided in our problem setup. Considering that a Markovian cost function is not applicable to all applications, we propose to learn trajectory-wise safety signals by proposing SafetyTransformer. Besides, considering the case that some trajectories may resemble the safe demonstrations due to high returns but belong to unsafe behavior patterns, we propose to learn the discriminator in a return-agnostic manner.

**Architecture of SafetyTransformer.** To aggregate comprehensive trajectory information and derive safety signals, we utilize the GPT transformer architecture for our discriminator, capitalizing on its effectiveness in processing sequential data. Its causally masked self-attention mechanism is adept at capturing the non-Markovian features of trajectories. As shown in Figure 2, we generate dual input embeddings for each state-action pair in the trajectory. These embeddings are fed into the causal transformer network, producing a series of embeddings , where each output embedding $\mathbf{x}_t$ is influenced solely by the preceding and current input embeddings up to $t$.

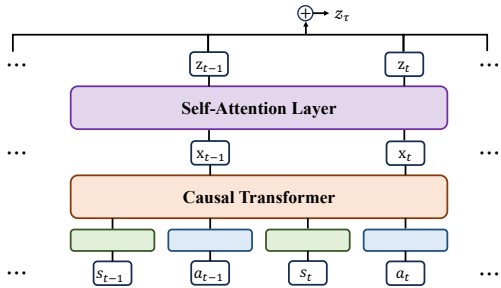

Figure 2: The network architecture of the SafetyTransformer.

To further capture the features of critical state-action pairs, we additionally employ a self-attention layer after obtaining the embeddings $\{\mathbf{x}_t\}_{t=1}^T$, similar to the practice in Preference-Transformer (Kim et al., 2022). For each $\mathbf{x}_t$, we use separate networks to obtain the mappings of key $\mathbf{k}_t \in \mathbb{R}^d$, query $\mathbf{q}_t \in \mathbb{R}^d$, and value $v_t \in \mathbb{R}$, where $d$ denotes the vector dimension. The mappings are computed:

$$\mathbf{k}_t = \mathbf{W}_k \mathbf{x}_t + \mathbf{b}_k, \quad \mathbf{q}_t = \mathbf{W}_q \mathbf{x}_t + \mathbf{b}_q, \quad v_t = \mathbf{w}_v^\top \mathbf{x}_t + b_v,$$

where $\mathbf{W}_k$, $\mathbf{W}_q$ are weight matrices, $\mathbf{b}_k$, $\mathbf{b}_q$ are bias vectors, $\mathbf{w}_v$ is a weight vector, and $b_v$ is a scalar bias. Next, we can obtain a series of vectors $\mathbf{z}_t$ through attention mechanism:

$$\mathbf{z}_t = \sum_{s=1}^T \alpha_{ts} v_s, \text{ where } \alpha_{ts} = \frac{\exp(\mathbf{q}_t^\top \mathbf{k}_s)}{\sum_{j=1}^T \exp(\mathbf{q}_t^\top \mathbf{k}_j)}.$$

Finally, we average all $\mathbf{z}_t$ values and apply the Sigmoid activation function to derive the safety signal, expressed as $z_\tau = \texttt{Sigmoid}\left(\frac{1}{T}\sum_{t=1}^T \mathbf{z}_t\right)$. We define the SafetyTransformer as $D_\phi$, where $\phi$ represents the model parameters. In the following context, $D_\phi(\tau)$ is equivalent to $z_\tau$.

**Return-agnostic learning.** To equip the transformer with the capability of expressing appropriate safety signals, we propose to utilize it as a trajectory-wise discriminator, like the practice in IRL, and train it utilizing the given safe demonstrations and the supplementary offline dataset. More

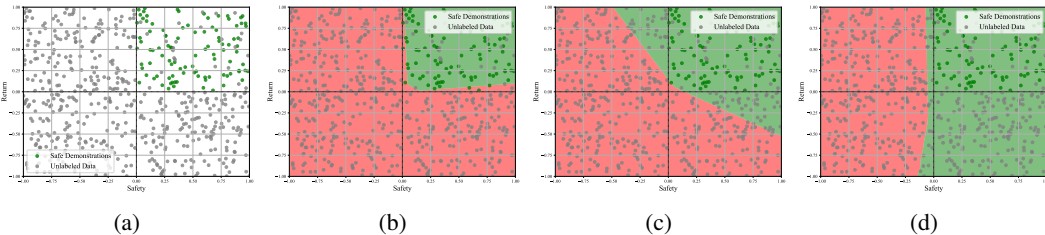

|  (a)  |  (b)  |  (c)  |  (d)  |

Figure 3: Illustrative experiment on a simple 2D environment. Figure (a) visualizes the safe demonstrations and unlabeled data. Figures (b) and (c) show the results without return-agnostic learning. The difference is that (c) uses gradient penalty to avoid overfitting. (d) is the result obtained applying return-agnostic learning. The results show our method recalls more safe samples, leading to an improved decision boundary.

specifically, our SafetyTransformer outputs the logit of one trajectory to be safe. The classical discriminator loss is typically defined as:

$$L_D(\phi) = -\mathbb{E}_{\tau^S \sim \mathcal{D}^S}[\log D_\phi(\tau^S)] - \mathbb{E}_{\tau^U \sim \mathcal{D}^U}[\log(1 - D_\phi(\tau^U)], \tag{1}$$

where we utilize $D_\phi$ to indicate the discriminator network parameterized by $\phi$.

However, our problem scenario significantly differs from the previous works since the quality of the trajectories is not only determined by the task return. Instead, it resembles the multi-objective problem and each trajectory contains the metrics of both task return and safety. For example, in path navigation task, the agent may need to consider both the path length (task return) and the collision risk (safety). It can burden challenges on the discriminator training because some trajectories may resemble the demonstrations due to high returns, but actually they belong to unsafe behavior patterns.

We take a simple 2D case in Figure 3a as an example. In this case, each sample is evaluated in two dimensions: return and safety. The safe demonstrations are safe samples with high return. Directly optimizing objective in Equation (1) leads to the decision boundary in Figure 3b, which fails to effectively recall the safe samples. When we utilize techniques like gradient penalty to improve the generalization ability of the discriminator in Figure 3c, it mistakenly assigns high weights to a considerable number of unsafe high return samples since their similarities to safe demonstrations.

To alleviate this issue, the core is to make the discriminator focus more on the safety factors of the trajectories, instead of factors relating to the return. To achieve this goal, we propose to learn a return-agnostic discriminator by minimizing the mutual information (MI) between the discriminator output $z_\tau$ and the trajectory return $r_\tau$. The MI between $r_\tau$ and $z_\tau$ is:

$$I(z_\tau; r_\tau) = \mathbb{E}_{p(z_\tau, r_\tau)} \left[\log p(r_\tau | z_\tau)\right] - \mathbb{E}_{p(r_\tau)} \left[\log p(r_\tau)\right]. \tag{2}$$

Since directly minimizing this MI term is intractable, we propose to minimize the upper bound of $I(z_\tau; r_\tau)$ instead. More specifically, we choose to minimize a variational contrastive log-ratio upper bound (Cheng et al., 2020), which is explained in the following theorem.

**Theorem 1** (Variational Contrastive Log-Ratio Upper Bound (Cheng et al., 2020)). *Let $q(r_\tau \mid z_\tau; \theta)$ be a variational approximation of $p(r_\tau \mid z_\tau)$ with parameter $\theta$. Denote $q(z_\tau, r_\tau; \theta) = q(r_\tau \mid z_\tau; \theta)p(z_\tau)$. If the following condition is met:*

$$D_{\mathrm{KL}}(p(z_\tau, r_\tau) \| q(z_\tau, r_\tau; \theta)) \leq D_{\mathrm{KL}}(p(r_\tau)p(z_\tau) \| q(z_\tau, r_\tau; \theta)), \tag{3}$$

*then the mutual information $I(z_\tau; r_\tau)$ is bounded above by $I_{\mathrm{vCLUB}}(z_\tau; r_\tau)$, where:*

$$I_{\mathrm{vCLUB}}(z_\tau; r_\tau) = \mathbb{E}_{p(z_\tau, r_\tau)} \left[\log q(r_\tau \mid z_\tau; \theta)\right] - \mathbb{E}_{p(z_\tau)}\mathbb{E}_{p(r_\tau)} \left[\log q(r_\tau \mid z_\tau; \theta)\right]. \tag{4}$$

The proof can be found in Appendix A. This theorem reveals that we can minimize $I(z_\tau; r_\tau)$ by optimizing the surrogate loss $I_{\mathrm{vCLUB}}(z_\tau; r_\tau)$. This surrogate loss can be intuitively understood as aiming to optimize $z_\tau$ to hinder $q_\theta$ from predicting the corresponding reward $r_\tau$, while encouraging it to predict incorrect rewards.

Meanwhile, this upper bound holds only if Equation (3) is met. Therefore, in practice we also need to optimize $\theta$ to minimize the term $D_{\mathrm{KL}}(p(z_\tau, r_\tau) \| q(z_\tau, r_\tau; \theta))$, in order to satisfy Equation (3).

According to Equation (5), we know that minimizing $D_{\mathrm{KL}}(p(z_\tau, r_\tau) \| q(z_\tau, r_\tau; \theta))$ is equivalent to maximizing $\mathbb{E}_{p(z_\tau)} \mathbb{E}_{p(r_\tau)} [\log q(r_\tau \mid z_\tau; \theta)]$, which corresponds to the data's log-likelihood.

$$
\begin{aligned}
D_{\mathrm{KL}}(p(z_\tau, r_\tau) \| q(z_\tau, r_\tau; \theta)) &= \mathbb{E}_{p(z_\tau, r_\tau)} [\log p(z_\tau, r_\tau)] - \mathbb{E}_{p(z_\tau, r_\tau)} [\log q(z_\tau, r_\tau; \theta)] \\
&= \mathbb{E}_{p(z_\tau, r_\tau)} [\log p(r_\tau \mid z_\tau)] - \mathbb{E}_{p(z_\tau, r_\tau)} [\log q(r_\tau \mid z_\tau; \theta)].
\end{aligned}
\tag{5}
$$

Furthermore, to stabilize the training process in practical implementation, we discretize the reward space and model $q(r_\tau \mid z_\tau; \theta)$ as a classifier. More implementation details can be found in Appendix D. By adding this return-agnostic learning loss, we force our SafetyTransformer to ignore information related to return when outputting $z_\tau$, so that $z_\tau$ contains more information concerning safety factors. In the 2D example shown in Figure 3, our method obtains a more accurate decision boundary.

## 3.2 DIFFUSION-GUIDED SAFE POLICY

Benefiting from the learned discriminator in Section 3.1, we can label each trajectory with one safety signal that denotes its weight of safety. Next, we aim to leverage both the reward and safety information in the offline dataset to learn a safe policy with high return. It is non-trivial as it corresponds to a multi-objective optimization problem solely using offline dataset. In this process, one core question is *how the trajectory distribution changes according to the task return and safety constraint*? Inspired by the remarkable capability of Diffusion Model (Ho et al., 2020b) to express distributions in recent applications (Lugmayr et al., 2022; Luo & Hu, 2021; Croitoru et al., 2023; Li et al., 2022c), we propose to utilize a conditional diffusion model to capture the relationship between trajectory return, safety signal, and the trajectory distribution.

To be concrete, since there can exist significant differences among the trajectory distributions corresponding to varying trajectory returns and safety constraints, we utilize a conditional diffusion model to generate trajectories conditioned on trajectory return $r_\tau$ and safety signal $z_\tau$. It is specifically implemented by a classifier-free diffusion model, which means that $r_\tau$ and $z_\tau$ are utilized as conditional inputs for each step in the diffusion's reverse denoising process. The training loss is defined as:

$$
L_{\mathrm{Diff}}(\psi) = \mathbb{E}_{t, x^0, \psi, r, z} \left[ \| \hat{\epsilon}_t(x^t, t, r, z; \psi) - \epsilon \|^2 \right],
\tag{6}
$$

where $x^0$ indicates the trajectory $\tau$, and $r, z$ are corresponding trajectory return and safety signal. More detailed background knowledge of diffusion models can be found in Appendix B.

The trained diffusion model enables us to derive trajectory distributions under different inputs of trajectory returns and safety constraints. Thus, by inputting high returns and safety weights, we can obtain the desirable trajectory distribution. However, this alone is not sufficient to support a complete decision-making process, as it only provides the joint distribution of state and action sequences, instead of the desirable action distribution when given the states. Thus, to further optimize a safe policy that can be deployed, we utilize the diffusion model to generate trajectories of both high return and safety signal, denoted as $\mathcal{D}^G$. These generated trajectories, together with the safe demonstrations, are employed to optimize a safe policy $\pi_\eta$ by optimizing the BC loss $L_{\mathrm{BC}}(\eta) = \mathbb{E}_{(s,a) \sim \mathcal{D}^S \cup \mathcal{D}^G} \left[ (\pi_\eta(s) - a)^2 \right]$.

## 4 EXPERIMENTS

**Experiment Setup.** To evaluate the effectiveness of various algorithms under our proposed new problem setup, we first construct an offline dataset suite based on popular safe RL benchmark environments, including tasks from SafetyGym, BulletGym, and MetaDrive. Within this offline dataset suite, each task is equipped with an offline dataset without any cost labels and $10 \sim 15$ safe demonstration trajectories. In this section, we primarily conduct experiments on this dataset suite, aiming to evaluate the performance of some existing algorithms on this new problem setup and validate whether our proposed algorithm can solve this problem well. We hope this dataset can facilitate further future research concerning this direction. More details about the offline datasets are provided in Appendix C.3.

**Baseline Design.** To investigate the features of our new proposed problem setup, and validate the effectiveness of our proposed algorithm, we mainly incorporate the following baselines for evaluation:

Table 1: Performance of algorithms across different tasks. Each result is averaged over 20 episodes and 5 random seeds. The unsafe results are marked gray, while safe results are **bolded**. Among them, the highest safe result for each task is marked **blue**. When ranking the algorithms, safe results are ranked higher. If two results are both safe or unsafe, those with higher returns are ranked higher.

| Task | BC-All | BC-Safe | TD3+BC | IQL | CQL | DWBC | RGM | CDT-V | DSPO (Ours) |
|---|---|---|---|---|---|---|---|---|---|
| CarButton1 | 0.11±0.07 | 0.20±0.02 | 0.30±0.14 | 0.17±0.03 | 0.06±0.03 | 0.25±0.01 | 0.06±0.01 | 0.17±0.01 | **0.05±0.01** |
| CarButton2 | 0.03±0.04 | 0.16±0.06 | 0.44±0.02 | 0.07±0.01 | -0.05±0.02 | 0.17±0.07 | -0.20±0.04 | 0.38±0.01 | **0.04±0.01** |
| CarGoal1 | **0.37±0.04** | 0.59±0.02 | 0.65±0.02 | **0.38±0.01** | 0.23±0.02 | 0.63±0.02 | 0.36±0.01 | 0.69±0.01 | **0.42±0.05** |
| CarGoal2 | 0.25±0.04 | 0.45±0.09 | 0.69±0.02 | 0.26±0.02 | **0.16±0.04** | 0.57±0.01 | 0.26±0.04 | 0.65±0.01 | **0.21±0.01** |
| CarPush1 | **0.16±0.03** | 0.28±0.02 | 0.25±0.02 | **0.19±0.02** | **0.19±0.02** | 0.30±0.01 | **0.19±0.01** | 0.31±0.01 | **0.19±0.03** |
| CarPush2 | 0.07±0.06 | 0.15±0.03 | 0.18±0.03 | 0.06±0.01 | 0.06±0.02 | 0.19±0.02 | 0.08±0.01 | 0.19±0.01 | **0.05±0.01** |
| PointButton1 | 0.15±0.04 | 0.48±0.02 | 0.31±0.13 | 0.23±0.02 | 0.07±0.01 | 0.51±0.03 | 0.18±0.01 | 0.62±0.01 | **0.08±0.01** |
| PointButton2 | 0.23±0.07 | 0.38±0.06 | 0.57±0.04 | 0.29±0.01 | 0.13±0.01 | 0.46±0.03 | 0.25±0.01 | 0.60±0.01 | **0.10±0.01** |
| PointGoal1 | 0.55±0.09 | 0.50±0.11 | 0.75±0.01 | 0.60±0.01 | 0.46±0.04 | 0.71±0.01 | 0.48±0.02 | 0.72±0.02 | **0.40±0.03** |
| PointGoal2 | 0.49±0.08 | 0.39±0.06 | 0.80±0.01 | 0.55±0.03 | 0.35±0.02 | 0.63±0.02 | 0.51±0.03 | 0.73±0.01 | **0.31±0.02** |
| PointPush1 | **0.20±0.05** | 0.23±0.06 | 0.34±0.02 | **0.18±0.01** | 0.14±0.03 | 0.25±0.01 | 0.18±0.01 | 0.31±0.01 | **0.18±0.01** |
| PointPush2 | 0.15±0.04 | 0.19±0.03 | 0.20±0.03 | 0.14±0.01 | **0.10±0.01** | 0.04±0.01 | 0.10±0.04 | 0.24±0.02 | **0.24±0.01** |
| HalfCheetahVel | 0.96±0.02 | **0.70±0.21** | 1.08±0.01 | 1.14±0.01 | 1.03±0.02 | 1.04±0.01 | **0.45±0.03** | 0.99±0.01 | **0.85±0.02** |
| SwimmerVel | 0.55±0.09 | **0.67±0.01** | 0.29±0.03 | 0.38±0.05 | **0.05±0.01** | 0.68±0.01 | 0.18±0.13 | 0.68±0.01 | **0.68±0.01** |
| Walker2dVel | **0.78±0.02** | **0.56±0.19** | 0.94±0.02 | 0.95±0.02 | 0.78±0.18 | **0.79±0.01** | **0.79±0.01** | 0.79±0.01 | **0.79±0.01** |
| **SafetyGym Average Rank** | 6.1 | 5.4 | 4.3 | 5.1 | 6.1 | 4.7 | 6.4 | 4.2 | **1.1** |
| AntCircle | 0.62±0.03 | 0.22±0.02 | 0.94±0.03 | 0.90±0.01 | **0.00±0.00** | 0.23±0.01 | 0.56±0.04 | 0.61±0.01 | 0.39±0.02 |
| AntRun | 0.72±0.06 | **0.12±0.01** | 0.31±0.12 | 0.77±0.01 | 0.79±0.01 | **0.48±0.01** | 0.65±0.01 | 0.76±0.01 | **0.69±0.02** |
| CarCircle | 0.71±0.02 | 0.67±0.05 | 0.88±0.01 | 0.91±0.01 | 0.20±0.02 | 0.72±0.01 | 0.80±0.01 | 0.92±0.01 | **0.73±0.01** |
| DroneCircle | 0.64±0.02 | 0.15±0.12 | 0.65±0.01 | 0.64±0.02 | 0.17±0.10 | 0.32±0.06 | 0.67±0.01 | 0.74±0.01 | **0.57±0.03** |
| DroneRun | 0.56±0.04 | 0.58±0.01 | 0.75±0.08 | 0.65±0.03 | -0.05±0.06 | 0.60±0.02 | **0.56±0.01** | 0.36±0.01 | **0.57±0.02** |
| **BulletGym Average Rank** | 6.0 | 7.0 | 4.4 | 4.0 | 6.2 | 5.6 | 4.8 | 4.6 | **2.2** |
| easydense | **0.19±0.29** | **0.15±0.14** | 0.88±0.01 | 0.68±0.05 | 0.18±0.05 | **0.60±0.01** | 0.48±0.01 | 0.58±0.08 | **0.20±0.01** |
| mediummean | **0.23±0.30** | **0.24±0.19** | 0.96±0.01 | 0.85±0.02 | 0.14±0.13 | 0.71±0.07 | 0.90±0.01 | **0.24±0.01** | **0.24±0.02** |
| hardsparse | 0.48±0.06 | 0.07±0.05 | 0.50±0.01 | 0.13±0.02 | **0.08±0.06** | **0.45±0.01** | 0.51±0.02 | 0.23±0.01 | **0.10±0.01** |
| **MetaDrive Average Rank** | 4.7 | 5.0 | 5.7 | 7.6 | 5.7 | 3.7 | 4.3 | 5.3 | **2.0** |

- **BC-All**: This baseline learns policies through behavior cloning utilizing all offline data, i.e., the trajectories from both safe demonstrations and the supplementary offline dataset.

- **BC-Safe**: This baseline also conducts behavior cloning to optimize policies, but utilizing the safe demonstrations only.

- **Offline RL Methods**: We include three offline RL methods, i.e., TD3+BC (Fujimoto & Gu, 2021), IQL (Kostrikov et al., 2021), and CQL (Kumar et al., 2020), that learn the policies based on the reward signals within the offline dataset.

- **DWBC** (Xu et al., 2022a): DWBC is an offline imitation learning (IL) method that learns from mixed-quality data. The safe demonstrations and offline supplementary dataset in our experiments respectively correspond to the limited expert data and mixed-quality dataset in the setting of DWBC method.

- **RGM** (Li et al., 2022a): RGM is an offline policy optimization method that learns from the offline dataset with imperfect rewards. It proposes a bi-level optimization framework to correct the rewards, enabling the resulting policy to match the limited expert data. Considering the task reward labels within the dataset as imperfect rewards, RGM can be naturally applied to our problem setting.

- **CDT-V (CDT-Variant)**: CDT (Liu et al., 2023) is one of the SOTA offline safe RL algorithms, which proposes a constraint decision transformer for safe decision-making. However, it relies on Markovian cost labels within the dataset. To incorporate it for comparison, we design a variant of CDT, CDT-V, that first learns transition-level cost signals and then applies the CDT algorithm.

In this section, we mainly aim to utilize the experimental results to answer the following research questions: 1) How do various algorithms perform under our problem setup, and can our method achieve superior safety performance over different categories of algorithms (Section 4.1)? 2) Does our designed SafetyTransformer architecture benefit the discriminator learning (Section 4.2)? 3) How does the return-agnostic loss affect the learning results of safety signals (Section 4.3)?

## 4.1 SAFETY PERFORMANCE COMPARISON

We first test all the baselines on the offline dataset suite to evaluate their performance when learning from the offline dataset without cost labels. The complete evaluation results are provided in Table 1. Although the problem setting resembles some real-world task scenarios, it is also more challenging as shown in the results in Table 1, where the majority of these baselines struggle to achieve safe policies in most scenarios. For example, one offline RL method TD3+BC, which applies the TD3 (Fujimoto et al., 2018) algorithm on the offline dataset with BC regularization, fails to obtain safe policies across all tasks. This reveals that directly applying existing offline RL methods in real-world applications can involve significant risks, calling for more safe and effective solutions.

Despite not failing to obtain safe policies across all tasks like TD3+BC, the other two offline RL methods, IQL and CQL, also face great challenges in obtaining safe policies in most scenarios. This phenomenon motivates our approach which involves first learning the safety signals for trajectories to guide the safe policy learning, rather than solely exploiting the reward signals within the offline dataset. The two behavior cloning methods, BC-All and BC-Safe, ignore the reward information in the dataset, but still fail to achieve good safety performance. The reason is that for BC-All, the training dataset is complex and mixed, containing various trajectory data, resulting in no guaranteed learning outcomes. For BC-Safe, the safe demonstrations are limited, which may strengthen the compounding error issue of BC method. The other three baselines, despite incorporating additional algorithm designs, also face poor safety performance in most tasks. In contrast, by leveraging trajectory-wise safety signal learning and diffusion-guided policy optimization, our method consistently achieves safe policies with competitive scores across most tasks.

**Effect of return-agnostic discriminator.** In fact, DWBC shares similarities with our approach in that they both derive the final policy through BC and both learn a discriminator. However, DWBC ignores the bi-objective (i.e., task performance & safety) property of the offline dataset and directly follows a target similar to Equation (1) to train the discriminator network. The issue with this practice is that some unsafe but high-return trajectories in the dataset might also resemble the safe demonstrations, causing the discriminator to assign large weight to these trajectories as well. This explains why DWBC exhibits high task performance in many tasks but neglects safety requirements.

**Transition-wise v.s. Trajectory-wise safety signal.** The RGM and CDT-V algorithms incorporate safety into consideration by treating task reward information as imperfect and correcting it accordingly via a bi-level optimization method. Essentially, they both learn safety signals at the transition level. However, they still fail to achieve safe results in quite many tasks. We hypothesize one possible reason is that the transition-level safety signals are hard to reconstruct and inaccurate safety signals can lead to unfavorable policy results. In contrast, the better safety performance of our approach, to some extent, demonstrates the superiority of our practice.

## 4.2 TRANSFORMER ARCHITECTURE HELPS CAPTURE TRAJECTORY FEATURES

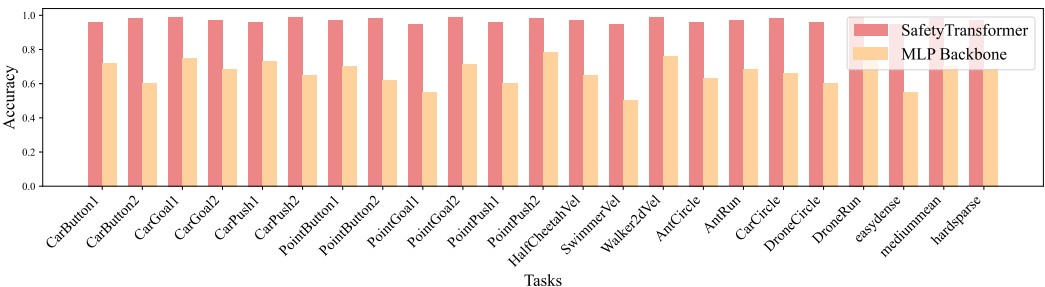

Figure 4: Ablation study to compare SafetyTransformer architecture with MLP backbone.

Since we aim to learn trajectory-wise safety signals, we propose a transformer-architecture discriminator, SafetyTransformer, to help extract the non-Markovian trajectory features. To validate the effectiveness of this architecture design, we compare it with a Multi-Layer Perceptron (MLP) backbone with similar parameter size. Specifically, we separate the safe trajectories and unsafe

Table 2: Ablation results on return-agnostic learning. Pear Corr. is the short for Pearson correlation coefficient, which represents the correlation between the weights output by the discriminator and whether the trajectory is actually safe. "gp" means utilizing gradient-penalty. The best result in each metric is highlighted in **bold**, and the gray values in the Final Score indicate that the agent fails to achieve safe behavior.

| Method | Recall % | Accuracy % | F1 Score % | Pearson Corr. | Final Score |
|---|---|---|---|---|---|
| w/ return-agnostic (ours) | **95.28±0.01** | **89.28±0.02** | **89.52±0.01** | **0.84±0.01** | **48.68±4.87** |
| w/o return-agnostic | 6.01±0.04 | 53.61±0.05 | 11.07±0.04 | 0.10±0.01 | 34.00±14.60 |
| w/o return-agnostic, w/ gp | 83.69±0.02 | 74.43±0.03 | 75.88±0.02 | 0.51±0.02 | 68.15±4.87 |

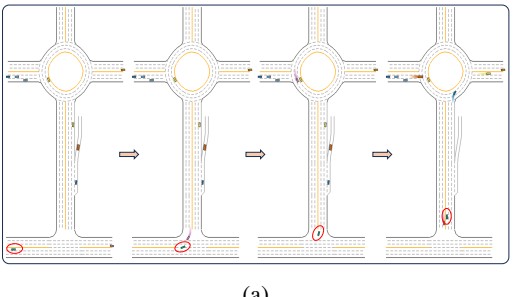

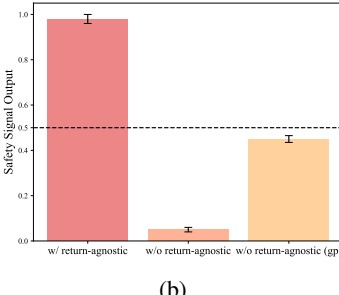

|  (a)  |  (b)  |
|---|---|

Figure 5: A specific case study on MetaDrive-hardsparse. Only our method applying return-agnostic learning determine the sample as safe and successfully recall it.

trajectories for each task. This two architecture backbones are trained for this classification task, and then evaluated on a separate part of leave-out data. The final evaluation results presented in Figure 4 reveal that our SafetyTransformer architecture significantly outperforms the MLP backbone. For example, on SwimmerVel, our SafetyTransformer is nearly twice the accuracy of MLP backbone.

## 4.3 Return-agnostic Learning Brings Better Safety Signals

In our approach, to avoid the discriminator assigning large weight to unsafe trajectories yet with high returns, we propose a return-agnostic learning method that aims to minimize the MI between discriminator output and the trajectory return. In this section, we ablate it on the hardsparse task of MetaDrive to investigate its effect on the discriminator learning and final policy performance.

Specifically, we proposed two ablation baselines: one that trains the discriminator directly using safe demonstrations and unlabeled data (w/o return-agnostic), and another that incorporates gradient penalty on this basis to prevent overfitting (w/o return-agnostic, w/ gp). We evaluate various metrics, including the proportion of safe samples successfully recalled from unlabeled data (**Recall**), the overall accuracy (**Accuracy**) of safety determination and its F1 score (**F1 Score**), the Pearson correlation coefficient between the discriminator output weight and actual trajectory safety (**Pearson Corr.**), as well as the performance of the final derived policy (**Final Score**). The results in Table 2 show that our method consistently achieves best performance across different metrics when applying return-agnostic learning, demonstrating its effectiveness.

To enhance understanding, we present a specific case study in Figure 5. In this trajectory, the vehicle successfully avoids collisions and stays within the lane boundaries, making it a safe trajectory sample that embodies safe decision-making knowledge. However, it has a low return due to the vehicle's slow speed. Directly training the discriminator without return-agnostic learning fails to assign a significant weight to this sample, which aligns with the low recall ratio in Table 2. While incorporating gradient penalty mitigates this issue, it still does not recall this case. Furthermore, it is likely to recall some high-return unsafe samples, which explains why it achieves a high final policy score yet being unsafe.

## 5 Related Work

### 5.1 Offline Safe Reinforcement Learning

The goal of offline safe reinforcement learning is to learn safe RL policies from offline datasets, addressing the safety concerns that arise during the training and deployment phases of conventional

RL (Le et al., 2019; Xu et al., 2022b; Lee et al., 2022; Liu et al., 2023; Lin et al., 2023). Offline safe RL challenges the conventional issues of distribution shifts between data-collection and learned policies (Fujimoto et al., 2019; Fujimoto & Gu, 2021; Kostrikov et al., 2021), and confronts additional complexities due to cost constraints. To address these issues, previous works have made significant efforts. CBPL (Le et al., 2019) approaches offline safe RL as a problem of batch policy learning under constraints and addresses it from a game-theoretic perspective, employing Fitted Q Evaluation (FQE) for policy evaluation and Fitted Q Iteration (FQI) for policy improvement. CPQ (Xu et al., 2022b) identifies both original unsafe actions and out-of-distribution actions as unsafe within the dataset, and modifies the Bellman update of the reward critic to penalize such unsafe state-action pairs. COptiDICE (Lee et al., 2022) adeptly mimics the behaviors within the dataset, adjusted by the distribution corrections of the optimal policy, while adhering to upper cost constraints. CDT (Liu et al., 2023) introduces a method based on a return-conditioned sequential modeling framework that incorporates safety constraints, enabling rapid adaptation of policies to varying deployment conditions. TREBI (Lin et al., 2023) revisits the issue of real-time budget constraints from the perspective of trajectory distribution and addresses it through diffusion model planning. However, these existing methods all require a step-wise Markovian cost function and cannot be directly applied to the setting studied in this paper, while our method DSPO stands out as the only method that can learn safe policies without cost labels.

## 5.2 DIFFUSION MODEL FOR DECISION MAKING

Recently, diffusion models, a class of generative models, have demonstrated remarkable performance across various fields, particularly excelling in text-to-image generation (Rombach et al., 2021; Saharia et al., 2022). These models have outperformed previous generative models in terms of generation quality and training stability (Lugmayr et al., 2022; Luo & Hu, 2021; Croitoru et al., 2023; Li et al., 2022c). Recognizing the potent generative abilities of diffusion models, an increasing number of researchers are now leveraging them to tackle challenges in the RL realm.

One approach within this research utilizes diffusion models as planners (Janner et al., 2022; Liang et al., 2023; Ni et al., 2023; Zhu et al., 2023a), which, unlike traditional model-based methods (Moerland et al., 2023; Luo et al., 2022), avoid the compounding error issue through a non-regressive planning scheme (Janner et al., 2022). In some instances, guided-sampling techniques are used to generate trajectories with higher returns or to meet specific constraints (Liang et al., 2023; Xiao et al., 2023). Additionally, there is research where diffusion models are employed directly as the policy (Zhu et al., 2023b). The ability of diffusion models to represent complex multi-modal distributions allows these diffusion-based policies to achieve superior performance in both offline RL (Wang et al., 2022) and imitation learning scenarios (Pearce et al., 2022; Reuss et al., 2023).

Beyond these applications, another significant use of diffusion models is as data synthesizer (Zhu et al., 2023b; Zhang et al., 2024; Jackson et al., 2024). Researchers have trained diffusion models on existing replay buffers to generate additional training samples for policy learning (Lu et al., 2023), as well as using these models for data augmentation in multi-task learning, leading to improved performance compared to existing methods (He et al., 2023). Our method, DSPO, primarily uses the diffusion model as a data synthesizer. However, it is conditionally dependent on two variables, i.e., trajectory return and safety signal, for data augmentation.

## 6 CONCLUSION

Safe decision-making is crucial for real-world applications, necessitating offline safe reinforcement learning (RL) to ensure safe policy training and deployment. However, existing offline safe RL methods typically assume the presence of Markovian cost labels in the dataset, which may not be true for many real-world tasks. To advance offline safe RL towards practical scenarios, we propose a new problem setup: learning a safe policy from an offline dataset without any cost labels, but with limited safe demonstrations included. In this paper, we introduce a suite of offline datasets across tasks from SafetyGym, BulletGym, and MetaDrive. We also propose a novel algorithm to address this more challenging problem. Extensive evaluations demonstrate the superiority of our approach in tackling this issue. In the future, we aim to extend our benchmark to more complex real-world tasks and consider other problem settings that closely align with real-world applications.

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

## A   THEOREM PROOF

**Theorem 1** (Variational Contrastive Log-Ratio Upper Bound (Cheng et al., 2020)). *Let $q(r_\tau \mid z_\tau; \theta)$ be a variational approximation of $p(r_\tau \mid z_\tau)$ with parameter $\theta$. Denote $q(z_\tau, r_\tau; \theta) = q(r_\tau \mid z_\tau; \theta)p(z_\tau)$. If the following condition is met:*

$$D_{\mathrm{KL}}(p(z_\tau, r_\tau)\|q(z_\tau, r_\tau; \theta)) \leq D_{\mathrm{KL}}(p(r_\tau)p(z_\tau)\|q(z_\tau, r_\tau; \theta)), \tag{7}$$

*then the mutual information $I(z_\tau; r_\tau)$ is bounded above by $I_{\mathrm{vCLUB}}(z_\tau; r_\tau)$, where:*

$$I_{\mathrm{vCLUB}}(z_\tau; r_\tau) = \mathbb{E}_{p(z_\tau, r_\tau)}\left[\log q(r_\tau \mid z_\tau; \theta)\right] - \mathbb{E}_{p(z_\tau)}\mathbb{E}_{p(r_\tau)}\left[\log q(r_\tau \mid z_\tau; \theta)\right]. \tag{8}$$

*Proof.* Consider the following expressions:

$$D_{\mathrm{KL}}(p(z_\tau, r_\tau)\|q(z_\tau, r_\tau; \theta)) = \mathbb{E}_{p(z_\tau, r_\tau)}\left[\log \frac{p(z_\tau, r_\tau)}{q(z_\tau, r_\tau; \theta)}\right],$$

$$D_{\mathrm{KL}}(p(r_\tau)p(z_\tau)\|q(z_\tau, r_\tau; \theta)) = \mathbb{E}_{p(z_\tau)}\mathbb{E}_{p(r_\tau)}\left[\log \frac{p(r_\tau)p(z_\tau)}{q(z_\tau, r_\tau; \theta)}\right].$$

This allows us to rewrite Eq. (7) as:

$$\mathbb{E}_{p(z_\tau, r_\tau)}\left[\log \frac{p(z_\tau, r_\tau)}{q(z_\tau, r_\tau; \theta)}\right] \leq \mathbb{E}_{p(z_\tau)}\mathbb{E}_{p(r_\tau)}\left[\log \frac{p(r_\tau)p(z_\tau)}{q(z_\tau, r_\tau; \theta)}\right]. \tag{9}$$

Beside, since we have $q(z_\tau, r_\tau; \theta)$ with $q(r_\tau \mid z_\tau; \theta)p(z_\tau)$, we can further have the following derivations by substituting it:

$$\mathbb{E}_{p(z_\tau, r_\tau)}\left[\log \frac{p(z_\tau, r_\tau)}{q(z_\tau, r_\tau; \theta)}\right] \leq \mathbb{E}_{p(z_\tau)}\mathbb{E}_{p(r_\tau)}\left[\log \frac{p(r_\tau)p(z_\tau)}{q(z_\tau, r_\tau; \theta)}\right],$$

$$\mathbb{E}_{p(z_\tau, r_\tau)}\left[\log \frac{p(r_\tau \mid z_\tau)p(z_\tau)}{q(r_\tau \mid z_\tau; \theta)p(z_\tau)}\right] \leq \mathbb{E}_{p(z_\tau)}\mathbb{E}_{p(r_\tau)}\left[\log \frac{p(r_\tau)p(z_\tau)}{q(r_\tau \mid z_\tau; \theta)p(z_\tau)}\right],$$

$$\mathbb{E}_{p(z_\tau, r_\tau)}\left[\log \frac{p(r_\tau \mid z_\tau)}{q(r_\tau \mid z_\tau; \theta)}\right] \leq \mathbb{E}_{p(z_\tau)}\mathbb{E}_{p(r_\tau)}\left[\log \frac{p(r_\tau)}{q(r_\tau \mid z_\tau; \theta)}\right].$$

It is further equivalent to:

$$\mathbb{E}_{p(z_\tau, r_\tau)}[\log p(r_\tau \mid z_\tau)] - \mathbb{E}_{p(z_\tau, r_\tau)}[\log q(r_\tau \mid z_\tau; \theta)]$$
$$\leq \mathbb{E}_{p(r_\tau)}[\log p(r_\tau)] - \mathbb{E}_{p(z_\tau)}\mathbb{E}_{p(r_\tau)}[\log q(r_\tau \mid z_\tau; \theta)],$$
$$\Rightarrow \mathbb{E}_{p(z_\tau, r_\tau)}[\log p(r_\tau \mid z_\tau)] - \mathbb{E}_{p(r_\tau)}[\log p(r_\tau)]$$
$$\leq \mathbb{E}_{p(z_\tau, r_\tau)}[\log q(r_\tau \mid z_\tau; \theta)] - \mathbb{E}_{p(z_\tau)}\mathbb{E}_{p(r_\tau)}[\log q(r_\tau \mid z_\tau; \theta)]. \tag{10}$$

Moreover, we know that the MI term is defined as:

$$I(z_\tau; r_\tau) = \mathbb{E}_{p(z_\tau, r_\tau)}[\log p(r_\tau \mid z_\tau)] - \mathbb{E}_{p(r_\tau)}[\log p(r_\tau)].$$

Thus according to Equation (10), we have:

$$I(z_\tau; r_\tau) = \mathbb{E}_{p(z_\tau, r_\tau)}[\log p(r_\tau \mid z_\tau)] - \mathbb{E}_{p(r_\tau)}[\log p(r_\tau)]$$
$$\leq \mathbb{E}_{p(z_\tau, r_\tau)}[\log q(r_\tau \mid z_\tau; \theta)] - \mathbb{E}_{p(z_\tau)}\mathbb{E}_{p(r_\tau)}[\log q(r_\tau \mid z_\tau; \theta)]$$
$$= I_{\mathrm{vCLUB}}(z_\tau; r_\tau).$$

$$\square$$

## B   ADDITIONAL BACKGROUND KNOWLEDGE ABOUT DIFFUSION MODEL

Diffusion models, introduced by (Sohl-Dickstein et al., 2015), simulate the thermodynamic process of diffusion to create new data. Recently, the Denoising Diffusion Probabilistic Model (DDPM)

framework (Ho et al., 2020a) has demonstrated exceptional generative performance, surpassing traditional models like Variational Autoencoders (VAEs) (Kingma & Welling, 2013) and Generative Adversarial Networks (GANs) (Goodfellow et al., 2020), and achieving notable success across various applications (Lugmayr et al., 2022; Luo & Hu, 2021; Croitoru et al., 2023; Li et al., 2022c).

At the heart of this framework is a multi-step denoising procedure that transforms random noise $x^T \sim \mathcal{N}(0, I)$ into realistic data $x^0$, essentially reversing the forward diffusion sequence $x^{0:T}$. In the forward process, each transition $q(x^t|x^{t-1})$ involves adding Gaussian noise with a scheduled variance $\beta^t$, described by:

$$x^t = \sqrt{\alpha^t}x^{t-1} + \sqrt{1 - \alpha^t}\epsilon^t,$$

where $\alpha^t = 1 - \beta^t$ and $\epsilon^t \sim \mathcal{N}(0, I)$. Consequently, $x^t$ follows a Gaussian distribution $\mathcal{N}(\sqrt{\alpha^t}x^{t-1}, 1 - \alpha^t)$. As $T$ approaches infinity, $\alpha^T$ converges to zero, ensuring $x^T$ is effectively random noise from a standard Gaussian distribution.

To generate real data from this noise, a denoising network $p_\psi(x^{t-1}|x^t)$ is used, which reverses the forward transition. This involves a network $\epsilon_\psi(x^t, t)$ that predicts the noise component $\epsilon^t$, allowing the mean of $x^{t-1}$ to be calculated as:

$$\mu_\psi(x^t, t) = \frac{1}{\sqrt{\alpha^t}} \left( x^t - \frac{\beta^t}{\sqrt{1 - \bar{\alpha}^t}}\epsilon_\psi(x^t, t) \right),$$

where $\bar{\alpha}^t = \prod_{i=0}^{t} \alpha^i$. This equation enables the step-by-step denoising of random noise to obtain real data $x^0$. The network $\epsilon_\psi$ is trained by minimizing the loss $L_{\text{Diff}}(\psi) = \mathbb{E}_{x^0, t, \epsilon^t} \left[ \|\epsilon^t - \epsilon_\psi(x^t, t)\| \right]$.

## C    MORE EXPERIMENTAL DETAILS

### C.1    INTRODUCTION TO THE EXPERIMENTAL ENVIRONMENTS

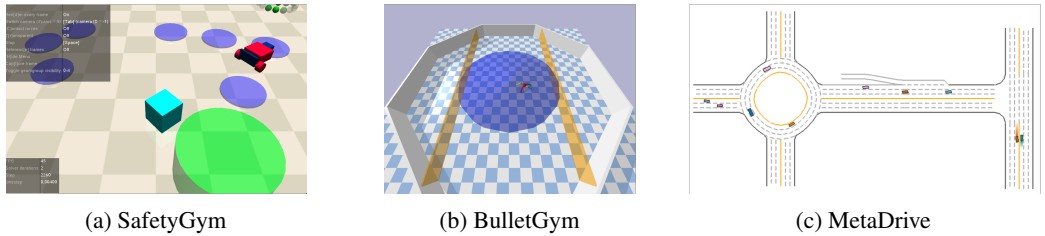

(a) SafetyGym                          (b) BulletGym                          (c) MetaDrive

Figure 6: There benchmark environments utilized in our experiments.

In this work, we primarily utilized three environments to conduct our offline safe RL experiments: **SafetyGym** (Ji et al., 2023), **BulletGym** (Gronauer, 2022), and **MetaDrive** (Li et al., 2022b). As shown in Figure 6, these environments focus on simulating real-world safety constraints, and have been popularly selected for exploring safe RL approaches, under both online and offline settings.

**SafetyGym**    SafetyGym is a benchmark developed by OpenAI for studying safe RL. It places agents in an environment together with different types of obstacles. The agents are expected to complete tasks (Goal, Button, Push) without violating specific safety rules, such as avoiding obstacles. The agents can be of different types, including Point, Car and Doggo (e.g., the Car in Figure 6a), and the task can be of different difficulty levels according to the density of obstacles.

**BulletGym**    BulletGym is a robust platform for simulating robotic systems, focusing on a range of locomotion and manipulation tasks. It is commonly utilized to investigate complex continuous control challenges. To facilitate benchmarking safe decision-making, BulletGym often imposes specific constraints on robotic velocities, ensuring that agents learn to navigate and act within safe operational limits.

**MetaDrive**    MetaDrive provides a highly customizable simulation environment designed for studying autonomous driving and multi-agent systems. It offers realistic driving scenarios where agents

must navigate complex road networks while making safe decisions, including avoiding collisions, staying within lane boundaries, and responding to dynamic traffic conditions. This allows for a robust training and testing environment for safe autonomous driving agents.

## C.2 MORE DETAILS ABOUT THE BASELINES

In Section 4, we provide a concise overview of the baseline designs utilized in our experiments. This includes offline RL methods, imitation learning techniques that incorporate supplementary data, a reward-correction method, and a variant of a SOTA offline safe RL approach. Here, we provide more details about the offline RL methods involved and some other categories of methods.

**TD3+BC.** TD3+BC (Fujimoto & Gu, 2021) is an enhancement of the TD3 (Dankwa & Zheng, 2019) algorithm, specifically designed for offline RL. It incorporates a behavior cloning (BC) term into the policy update process to regularize the learning, allowing the agent to leverage both the learned value function and demonstrations from a fixed dataset. This algorithm significantly simplifies the implementation compared to other offline RL methods, while achieving competitive performance.

**IQL.** IQL (Kostrikov et al., 2021) is a novel offline RL algorithm that avoids querying the values of unseen actions during training while still enabling multi-step dynamic programming updates. It achieves this through modifying the loss function of a standard SARSA-like temporal difference update, representing a significant advancement in offline RL in recent years.

**CQL.** CQL (Kumar et al., 2020) is a classic offline RL algorithm, which focuses on the value overestimation issue when learning from a static dataset utilizing the traditional Q-learning methods. By incorporating conservative estimates into the Q-function updates, CQL ensures that the learned policy remains robust and effective.

**DWBC.** DWBC (Xu et al., 2022a) is an offline imitation learning method designed to learn from datasets with a large proportion of suboptimal data. Similar to our problem setting, it assumes a small set of expert demonstrations and a large number of non-expert data. It solves the problem through enhancing behavior cloning by incorporating a discriminator that distinguishes between expert and non-expert data. The outputs of this discriminator are utilized as weights in the behavioral cloning loss, allowing the algorithm to selectively imitate high-quality transitions. Due to the similarity in problem structure, DWBC (Xu et al., 2022a) can be directly applied to our problem setting. We utilized the RGM's open-source implementation.

**RGM.** RGM (Li et al., 2022a) considers the problem setting where the reward can be imperfect, under an offline learning setting. Although the considered is entirely different from that of our work, the solution presented in RGM also assumes that there is a small amount of expert demonstrations. It formulates the problem as a bi-level optimization task. The upper layer adjusts a reward correction term to minimize the reward gap based on expert data, while the lower layer solves a pessimistic RL problem using the corrected rewards. This approach allows RGM to handle various types of imperfect rewards without needing online interactions. RGM can be naturally applied to our problem setting by considering that the task reward alone is imperfect reward, and there exist an oracle reward considering both performance and safety. We utilized their official open-source implementation.

**CTD-V.** CDT (Liu et al., 2023) (Constrained Decision Transformer) is an approach for offline safe reinforcement learning that addresses the trade-offs between safety and task performance. It dynamically adjusts constraint thresholds during deployment without the need for retraining. It leverages sequential modeling and introduces two key techniques: a stochastic policy with entropy regularization and data augmentation via return relabeling. These features help CDT learn adaptive, safe, and high-reward policies, outperforming existing baselines and enabling zero-shot adaptation to different constraints. However, both the training and testing phases of CDT requires transition-level cost labels, which is not accessible in our problem setting. To address it, our CDT-V learn the transition-level costs leveraging the approach in RGM. Actually, the difference between the corrected reward and the old reward can be viewed as an estimation of the safety at the transition level. We utilized the official open-source implementations of RGM and CDT.

Table 3: Some details about the experimental dataset.

| Task | Environment | Number of Safe Demonstrations | Number of Unlabled Data | Cost Limit |
|---|---|---|---|---|
| CarButton1 | SafetyGym | 15 | 800 | 20 |
| CarButton2 | SafetyGym | 15 | 800 | 20 |
| CarGoal1 | SafetyGym | 15 | 800 | 20 |
| CarGoal2 | SafetyGym | 15 | 800 | 20 |
| CarPush1 | SafetyGym | 15 | 800 | 20 |
| CarPush2 | SafetyGym | 15 | 800 | 20 |
| PointButton1 | SafetyGym | 15 | 800 | 20 |
| PointButton2 | SafetyGym | 15 | 800 | 20 |
| PointGoal1 | SafetyGym | 15 | 800 | 20 |
| PointGoal2 | SafetyGym | 15 | 800 | 20 |
| PointPush1 | SafetyGym | 15 | 800 | 20 |
| PointPush2 | SafetyGym | 15 | 800 | 20 |
| HalfCheetahVel | SafetyGym | 15 | 800 | 20 |
| SwimmerVel | SafetyGym | 15 | 800 | 20 |
| Walker2dVel | SafetyGym | 15 | 800 | 20 |
| AntCircle | BulletGym | 15 | 500 | 10 |
| AntRun | BulletGym | 15 | 500 | 10 |
| CarCircle | BulletGym | 15 | 500 | 10 |
| DroneCircle | BulletGym | 15 | 500 | 10 |
| DroneRun | BulletGym | 15 | 500 | 10 |
| easydense | MetaDrive | 15 | 400 | 10 |
| mediummean | MetaDrive | 15 | 400 | 10 |
| hardsparse | MetaDrive | 15 | 400 | 10 |

### C.3 DETAILS ABOUT THE OFFLINE DATASET

In this work, we build an offline dataset suite, where for each task there exist a few number of safe demonstrations and a supplementary offline dataset. Specifically, these datasets are collected on the environments of SafetyGym, BulletGym, and MetaDrive, leveraging the open-source tool FSRL (Liu et al., 2024). Regarding **the definition of trajectory safety**, we still rely on the cost information from these environments for judgment, even though this cost information is not accessible to the agent during training in our setup. When the cumulative cost of a trajectory exceeds a certain threshold, called cost limit, we determine that the trajectory is unsafe. More detailed information about our dataset is included in Table 3.

## D IMPLEMENTATION DETAILS

### D.1 MORE IMPLEMENTATION DETAILS OF DSPO

**Optimization of the variational contrastive log-ratio upper bound.** In order to make our discriminator be return-agnostic, thus avoiding assigning large safety weights to unsafe trajectories with high returns, we propose to add one return-agnostic loss term which is the variational contrastive log-ratio upper bound of the MI between trajectory return $r_\tau$ and safety signal $z_\tau$. To optimize this surrogate tractable loss, we need to parameterize $q(z_\tau, r_\tau; \theta)$. In practice, to stabilize the training process, we discretize the whole reward (scalar) space into ten bins, and model $q(z_\tau, r_\tau; \theta)$ as a ten-class classifier.

**Diffusion-based trajectory generation.** After obtaining the diffusion model, we utilize it to generate safe trajectories with high returns for behavior cloning. In this process, we need to feed conditional inputs to the diffusion model. In practice, for the input of trajectory return, we choose the value in the range of safe demonstrations; for the input of safety signal, we choose the smallest weight assigned to the safe demonstrations. The former ensures that the returns we input are within a reasonable distribution, while the latter aims to recall as diverse trajectories as possible under the condition of being safe to some extent.

---

**Algorithm 1** Training Algorithm

---

**Require:** Safe demonstrations $\mathcal{D}^S$, supplementary offline dataset $\mathcal{D}^U$
**Ensure:** Derived safe policy $\pi_\eta$.
 1: **Stage 1: Discriminator Training**
 2: **for** $i \in [1, 2, ..., \text{max\_iteration}]$ **do**
 3:    Train variational predictor $q_\theta$ to maximize the log-likelihood $\log q(z_\tau, r_\tau; \theta)$ by sampling trajectories from $\mathcal{D}^S \cup \mathcal{D}^D$.
 4:    Train discriminator $D_\phi$ to minimize $L_{\text{disc}}$ of Equation (1) and $I_{\text{vCLUB}}(z_\tau; r_\tau)$ of Equation (2) by sampling trajectories from $\mathcal{D}^S, \mathcal{D}^U$.
 5: **end for**
 6: Utilize the trained discriminator network to label each trajectory $\tau \in \mathcal{D}^S \cup \mathcal{D}^U$ with a safety signal $z_\tau$.
 7: **Stage 2: Policy Optimization**
 8: Train the conditional diffusion model to minimize the DDPM loss $L_{\text{Diff}}(\psi)$ of Equation (6).
 9: Utilize the trained diffusion model to generate a dataset of trajectories $\mathcal{D}^G$ by conditioning on high trajectory returns and safety signals.
10: Optimize agent policy $\pi_\eta$ by optimizing the BC loss $L_{\text{BC}}(\eta)$, with data sampling from $\mathcal{D}^S \cup \mathcal{D}^G$.

---

## D.2 OVERALL ALGORITHM FLOW

In this section, we introduce the overall procedure of our proposed approach DSPO, which is as introduced in Algorithm 1. In lines 1 to 6, we perform the first stage of our algorithm, where we train the SafetyTransformer network in a return-agnostic manner. Lines 7 to 10 introduce the second stage of our algorithm, where we firstly train a diffusion model conditioned on the trajectory return $r_\tau$ and safety signal $z_\tau$, and then generate trajectory dataset $\mathcal{D}^G$ of high returns and safety signals, which are utilized to behavior clone a safe policy $\pi_\eta$ together with $\mathcal{D}^S$.

## D.3 SELECTION OF HYPER-PARAMETERS

In this section, we provide the information of the main hyper-parameters of our approach in Table 4.

Table 4: Hyper-parameter configurations.

| Hyper-parameter | Value |
|---|---|
| Learning rate of the discriminator training | 0.00003 |
| Coefficient for the $L_{\text{vCLUB}}$ | 0.1 |
| Timesteps of training samples for underlying communication policy learning | $2M$ |
| Batch size for diffusion model training | 64 |
| Total iterations for diffusion model training | 200000 |
| Learning rate for diffusion model training | 0.0003 |
| Loss type for diffusion model training | $L_1$ loss |
| The iteration to start the EMA update | 1000 |
| Decay rate of the EMA update | 0.995 |
| Interval of the EMA update | 5 iterations |
| Diffusion timesteps, the steps for message denoising | 100 |
| Embedding dimension for the U-Net backbone of diffusion model | 64 |
| Hidden dimension for the attention layer of diffusion model | 128 |
| Batch size for policy behavior cloning | 64 |

