# OpenReview forum: "Diffusion-Guided Safe Policy Optimization From Cost-Label-Free Offline Dataset"
_ICLR.cc/2025/Conference — ICLR 2025 Conference Withdrawn Submission_

### Official Review · Reviewer_eCBV · 2024-10-16

**Soundness:** 2
**Presentation:** 2
**Contribution:** 1
**Rating:** 3
**Confidence:** 4

**Summary:**

This paper introduces DSPO, a method for offline safe reinforcement learning (RL) that learns safe policies without requiring explicit cost labels, using only a small set of safe demonstrations. Unlike prior approaches that depend on predefined Markovian cost functions, DSPO overcomes this limitation through a two-stage process: (1) training a discriminator to extract safety signals from trajectories, and (2) using a conditional diffusion model to generate high-return, safety-compliant trajectories, from which a policy is learned via behavior cloning. Experiments on SafetyGym, BulletGym, and MetaDrive show that DSPO achieves safer policies with better returns than the tested baselines.

**Strengths:**

1.  **Interesting topic:** The authors explore an engaging and important topic in offline RL—cost-free safe reinforcement learning. Advancing research in this direction holds significant potential for improving learning from safe demonstrations.

**Weaknesses:**

1. **Limited Contribution:** While the authors claim three major contributions—a more practical problem setup, a novel approach combining a safety-transformer with a diffusion model data generator, and the construction of an offline dataset suite—these contributions seem limited in significance.
   1.1. **Problem Setup:** Similar problem formulations, which assume the absence of ground-truth cost functions or rely on learning preferences from a few safe demonstrations, have been explored in prior works on inverse safe RL [1, 2] and meta-safe RL [3].
   1.2. **Methodological Novelty:** The safety-transformer has been widely explored in prior research [4], and classifier-free diffusion models in (safe) RL / RL dataset curation have also been extensively studied [5, 6, 7]. The authors do not sufficiently clarify how their approach differs from previous work, nor do they provide a comprehensive comparison with related diffusion model-based safe RL approaches, such as FISOR [9] and Safe Decision Diffuser [10].
   1.3. **Dataset Suite:** The proposed dataset suite is a modified version of the OSRL benchmark [8], created by selecting safe demonstrations and masking cost information.

2. **Vague Experimental Validation:** The main experimental results, presented in Table 1, focus only on reward returns. For safe RL tasks, it is crucial to also report detailed cost/safety metrics to ensure thorough validation.

3. **Additional Comments:** Please refer to my specific questions for further points.



Reference:

[1] Schlaginhaufen, Andreas, and Maryam Kamgarpour. "Identifiability and generalizability in constrained inverse reinforcement learning." In International Conference on Machine Learning, pp. 30224-30251. PMLR, 2023.

[2] Kim, Konwoo, Gokul Swamy, Zuxin Liu, Ding Zhao, Sanjiban Choudhury, and Steven Z. Wu. "Learning shared safety constraints from multi-task demonstrations." Advances in Neural Information Processing Systems 36 (2024).

[3] Lin, Qian, Zongkai Liu, Danying Mo, and Chao Yu. "An Offline Adaptation Framework for Constrained Multi-Objective Reinforcement Learning." arXiv preprint arXiv:2409.09958 (2024).

[4] Kim, Changyeon, Jongjin Park, Jinwoo Shin, Honglak Lee, Pieter Abbeel, and Kimin Lee. "Preference transformer: Modeling human preferences using transformers for rl." arXiv preprint arXiv:2303.00957 (2023).

[5] Lee, Jaewoo, Sujin Yun, Taeyoung Yun, and Jinkyoo Park. "GTA: Generative Trajectory Augmentation with Guidance for Offline Reinforcement Learning." arXiv preprint arXiv:2405.16907 (2024).

[6] Liang, Zhixuan, Yao Mu, Mingyu Ding, Fei Ni, Masayoshi Tomizuka, and Ping Luo. "Adaptdiffuser: Diffusion models as adaptive self-evolving planners." arXiv preprint arXiv:2302.01877 (2023).

[7] Yao, Yihang, Zhepeng Cen, Wenhao Ding, Haohong Lin, Shiqi Liu, Tingnan Zhang, Wenhao Yu, and Ding Zhao. "OASIS: Conditional Distribution Shaping for Offline Safe Reinforcement Learning." arXiv preprint arXiv:2407.14653 (2024).

[8] Liu, Zuxin, Zijian Guo, Haohong Lin, Yihang Yao, Jiacheng Zhu, Zhepeng Cen, Hanjiang Hu et al. "Datasets and benchmarks for offline safe reinforcement learning." arXiv preprint arXiv:2306.09303 (2023).

[9] Xiao, Wei, Tsun-Hsuan Wang, Chuang Gan, and Daniela Rus. "Safediffuser: Safe planning with diffusion probabilistic models." arXiv preprint arXiv:2306.00148 (2023).

[10] Zheng, Yinan, Jianxiong Li, Dongjie Yu, Yujie Yang, Shengbo Eben Li, Xianyuan Zhan, and Jingjing Liu. "Safe offline reinforcement learning with feasibility-guided diffusion model." arXiv preprint arXiv:2401.10700 (2024).

**Questions:**

1. **Line 169-170:** However, since the diffusion model cannot directly output decision actions, this raises a concern. Could you clarify this limitation?

2. **Figure 3a Example:** The authors used a simple 2D case in Figure 3a. This example appears too simplistic and therefore not convincing. Additionally, it contains far more safe demonstrations than those used in the main experiments. Could you provide visualization results for the actual experiment tasks?

3. **Comparison with Inverse RL and Other Baselines:** Have you compared your approach with inverse RL methods and other baselines? For instance, a simple baseline could involve learning a classifier for safe trajectories, labeling the remaining unlabeled dataset, and then using this dataset for offline safe RL training.

4. **Ground-Truth Cost Return:** Could you provide the ground-truth cost return for the evaluation results presented in Table 1, alongside the reward return?

5. **Omission of OSRL Tasks:** Why did you omit some tasks from the OSRL benchmark, such as car-run, ball-circle, and ball-run from Bullet Safety Gym? Can you share the full evaluation results?

6. **Diffusion Model Generator:** Are the trajectories generated by the diffusion model state-only trajectories or state-action pair trajectories?

7. **Dataset Size for Figure 4:** What is the size of the dataset used in the experiment shown in Figure 4?

---

### Official Review · Reviewer_4QT1 · 2024-10-17

**Soundness:** 1
**Presentation:** 3
**Contribution:** 2
**Rating:** 3
**Confidence:** 4

**Summary:**

This paper studies the problem of safe offline reinforcement learning without a **dense** Markovian cost function available, relying instead on a limited set of safe trajectories and extensive unlabeled data. This paper proposed a safety transformer architecture to produce trajectory-level safety signal trained on the labeled and unlabeled data. By doing so, the transformer can relabel the unlabeled data, allowing the subsquent training of a safety-conditional diffusion model that generates safe trajectories. To reduce the correlation between transformer prediction and returns, the authors proposed a return-agnostic learning objective to enforce the transformer do not focus on the returns, and thus can provide more accurate safety signals. The authors conducted experiments on the DSRL benchmark to demonstrate the effectiveness of the proposed method, and the design of the return-agnostic objective.

**Strengths:**

1. The introduction of the return-agnostic learning objective proves to be effective. I appreciate the intuitive illustration provided in Figure 3, which is exceptionally reader-friendly and quickly underscores the primary benefits of this design choice.
2. This paper is well-written, featuring clear motivations, a thorough clarification of challenges, and a distinct highlight of its contributions.
3. The figures and tables in this paper are exceptionally well-crafted.

**Weaknesses:**

However, several major weaknesses exists:

### **Motivations and paper problem setups**
-  **A dense Markovian cost function maybe not nessary for safe offline RL.** I agree with the author that designing a `dense` Markovian cost function that can capture the trajectory-level safety performance and can consider furture unsafe outcomes is pretty challenging. However, this does not imply that obtaining a `sparse` cost function is diffcult. Typically, we can straightforwardly annotate states as 0 or 1 to indicate whether they are in unsafe or safe regions, respectively.
    - While this simple sparse cost function may not offer trajectory-level safety evaluations, its associated Q-value or V-value function can propagate future information through bootstrapping. Thus, we can easily design a sparse cost function and derive its Q/V-value function to provide a trajectory-level safety signal.
    - This paper presupposes a safe dataset. From my perspective, there must be a straightforward safety judgment function used to determine the safety of this dataset. Consequently, this simple function could readily serve as the sparse cost function I previously mentioned.
    - The authors may believe that learning by sparse cost can be inefficient, since the TD signal is sparse. But in the offline setting, we can easily modify the sparse costs to sparse feasible value (1 for unsafe, -1 for safe)[1]. By doing so, learning an accurate Q/V value function becomes more easier.

### **Limited Experiments**

The authors can criticize that learning a Q value function through bootstrapping is unstable, and meanwhile argure that the proposed safety transformer offers more stable training. However, this kind of experiments are missed in the current manucript.
- **More popular safe offline RL methods should be evaluated**. The evaluated baselines in this paper include nearly no methods except CDT that specifically desinged for safe offline RL, probably because no costs are available. However, the authors should compare against them using sparse costs.
    - For example, the authors should compare other popular baselines, such as CPQ[2] and COptiDICE[3] based on the sparse cost function just as I mentioned, and the recent SOTA safe diffusion model FISOR[1] using sparse feasible function, etc.
    - The authors can also provide some intuitive illustration to demonstrate that the Q value learned by bootstrapping cannot offer accurate recall boundaries in Figure 3. This can further enhance the paper quality.

### **Theory: strong and strange assumption**
In theorem 1, the authors claim that they are minimizing an upper bound of the original intractable mutual information. However, this theorem holds on a pretty strong and strange condition.

- Specifically, the assumption is $D_{KL}(p(z,r)||q(z,r;\theta))\le D_{KL}(p(z)p(r)||q(z,r;\theta))$, where z is cost and r is return. The authors just directly claim that they are minizing an upper bound if this assumption holds. However, the authors never discuss when and why this assumption holds.
- I checked the proof of Theorem 1 and found that the authors just expand this assumption and then can derive the upper bound objective. So, this objective may not be the upper bound of the original mutual information anymore if this assumption does not hold. In this case, the authors may try to minimize a lower bound, which is incorrect.

[1] safe offline reinforcement learning with feasibility guided diffusion, ICLR 2024.

[2] Constraints Penalized Q-learning for Safe Offline Reinforcement Learning, AAAI 2022

[3] COptiDICE: Offline Constrained Reinforcement Learning via Stationary Distribution Correction Estimation, ICLR 2022.

[4] Safe Offline Reinforcement Learning with Real-Time Budget Constraints, ICML 2023

**Questions:**

See weaknesses for details.

---

### Official Review · Reviewer_5Ggq · 2024-11-03

**Soundness:** 2
**Presentation:** 2
**Contribution:** 2
**Rating:** 3
**Confidence:** 4

**Summary:**

This paper aims to learn a safe RL policy from a cost-label-free offline dataset. In this setting, the cost label is unavailable in most demonstrations while a small number of labeled safe demonstrations are also provided. The authors first train a transformer-based discriminator with safe and unlabeled demonstrations as positive and negative data respectively. Then they train a diffusion model based on the demonstrations with labels from discriminator to generate high-reward and safe data. The final policy is trained through behavioral cloning over the safe demonstrations and generated data. The experiments show that the proposed framework outperforms the baselines.

**Strengths:**

- This paper is clearly written and easy to follow.
- The experiment is extensive.

**Weaknesses:**

- The motivation of the new setup in this paper (i.e., the cost label is unavailable in most offline demonstration while a few safe demonstration are provided) is relatively weak: different from reward signal, the cost label in previous works is binary and is $+1$ only when there is an unsafe event (e.g., crash), which can be easily obtained or measured even in practical application. Moreover, even in the new setting, the criterion of safety is also determined by step-wise cost label as mentioned in line 949.
- The dataset used for experiment is built heavily based on previous work [1]. However, the authors have never cited it when introducing the dataset in main text (e.g., line 79, line 314), which is extremely improper and misleads the readers on its novelty. I urge the authors to clarify it.

[1] Datasets and benchmarks for offline safe reinforcement learning

**Questions:**

- In practice, how large the overlap between the safe and unlabeled demonstrations? In other word, how many unlabeled demonstrations are actually safe (but you regard it as unsafe in discriminator learning)?
- In experiment, why do you choose the number of safe demonstrations to be 15? Is there a parameter study of its influence on final performance?
- Why does the BC-All have safer performance than BC-safe on several tasks such as CarGoal1, CarPush1, PointPush1?
- Could you also report the cost performance in table 1?

---

### Official Review · Reviewer_51ay · 2024-11-04

**Soundness:** 2
**Presentation:** 2
**Contribution:** 2
**Rating:** 3
**Confidence:** 4

**Summary:**

The paper presents Diffusion-Guided Safe Policy Optimization (DSPO), a method for offline safe reinforcement learning (RL) that does not rely on cost labels in the dataset. Instead, DSPO uses a SafetyTransformer to infer safety signals from trajectories and subsequently employs a conditional diffusion model to generate safe trajectories that optimize a safe policy via behavior cloning (BC). Experiments on tasks in OSRL benchmarks indicate that DSPO outperforms baseline methods in achieving safe policies without cost labels.

**Strengths:**

- **Interesting perspective in data-centric Safe RL**: The paper explores challenging data-centric aspects of offline safe RL by attempting to learn safe policies without explicit cost labels, utilizing inferred safety signals and diffusion models for policy optimization.

- **Interesting motivation of reward agnostic cost discriminator**: it is somewhat insightful to mitigate the spurious correlation between return and cost output along the demonstration trajectories.

- **Thorough Experimentation**: comprehensive experiments across multiple environments and comparison with diverse baselines showcases the empirical advantage of DSPO.

**Weaknesses:**

Despite the extensive empirical results, there are quite a few weaknesses that significantly hinder the presentation of this paper:

- **Unrealistic assumption**: the authors claim they would like to push safe RL to more 'real-world applications', where most of the safety-critical scenarios are long-tailed (sparse cost). However, in their assumption in page 3, line 138-143, they assumed that a small number of safe demonstration and another supplementary offline dataset will be provided in offline safe RL problems. This seem to be controversial with the real-world setup that I just mentioned as it will be infeasible for any behavior policies to sample a large set of 'supplementary offline dataset' if they are essentially 'unsafe'.
- **Minor contribution in SafetyTransformer**: the authors mentioned about they would like to resolve the case where most of the trajectory label missing. However, in the line of works using inverse RL for constrained RL [1, 2], such question has been explored even under harder setting, yet these works were missing in the literature review.
- **Missing details in diffusion model**: the authors titled section 3.2 with 'DIFFUSION-GUIDED SAFE POLICY', however, they just used diffusion model to synthesize high-return trajectories and use bahavior cloning to learn the policy. Besides, the implementation of diffusion policies, e.g. the classifier-free guidance, is missing in the paper.
- **Uninformative theoretical results**: the theoretical results in Theorem 1 is essentially based on the non-negativity of the mutual information. This is extremely uninformative to the reader what DSPO benefits the existing offline safe RL methods, e.g., what is the benefits in the cost function.



> [1] Malik, Shehryar, et al. "Inverse constrained reinforcement learning." ICML 2021.
>
> [2] Subramanian, Sriram Ganapathi, et al. "Confidence aware inverse constrained reinforcement learning." ICML 2024.

**Questions:**

- **Justification of the assumption**: see the weakness part, could the authors elaborate more about how is the realism of such an assumption of offline dataset? And more specifically, could the authors provide more details about how they conducted their experiment in the OSRL benchmark to follow these assumptions?
- **About take-away of theoretical results**: in short words, what is the take-aways of Theorem 1, especially how would the authors interpret it with the key features, i.e. return and cost in the offline safe RL framework?

- **About diffusion model**:  Please clarify the following things:
  - What is the input of the BC-based policy, one-step state or sequence? What is the architecture of this BC policy?
  - Empirically, what is the impact of the mixture ratio between generated trajectories and true trajectories? Ablation studies may be helpful.

---

### Note · Authors · 2024-11-22

I have read and agree with the venue's withdrawal policy on behalf of myself and my co-authors.